# The Cell Autonomous and Non-Cell Autonomous Aspects of Neuronal Vulnerability and Resilience in Amyotrophic Lateral Sclerosis

**DOI:** 10.3390/biology11081191

**Published:** 2022-08-08

**Authors:** Christoph Schweingruber, Eva Hedlund

**Affiliations:** Department of Biochemistry and Biophysics, Stockholm University, Svante Arrhenius v. 16C, 106 91 Stockholm, Sweden

**Keywords:** amyotrophic lateral sclerosis, neurodegeneration, neurodegenerative cell metabolism, neuron

## Abstract

**Simple Summary:**

Amyotrophic lateral sclerosis (ALS) is a fatal disease characterized by a progressive paralysis due to the loss of particular neurons in our nervous system called motor neurons, that exert voluntary control of all our skeletal muscles. It is not entirely understood why motor neurons are particularly vulnerable in ALS, neither is it completely clear why certain groups of motor neurons, including those that regulate eye movement, are rather resilient to this disease. However, both vulnerability and resilience to ALS likely reflect cell intrinsic properties of different motor neuron subpopulations as well as non-cell autonomous events regulated by surrounding cell types. In this review we dissect the particular properties of different motor neuron types and their responses to disease that may underlie their respective vulnerabilities and resilience. Disease progression in ALS involves multiple cell types that are closely connected to motor neurons and we here also discuss their contributions to the differential vulnerability of motor neurons.

**Abstract:**

Amyotrophic lateral sclerosis (ALS) is defined by the loss of upper motor neurons (MNs) that project from the cerebral cortex to the brain stem and spinal cord and of lower MNs in the brain stem and spinal cord which innervate skeletal muscles, leading to spasticity, muscle atrophy, and paralysis. ALS involves several disease stages, and multiple cell types show dysfunction and play important roles during distinct phases of disease initiation and progression, subsequently leading to selective MN loss. Why MNs are particularly vulnerable in this lethal disease is still not entirely clear. Neither is it fully understood why certain MNs are more resilient to degeneration in ALS than others. Brain stem MNs of cranial nerves III, IV, and VI, which innervate our eye muscles, are highly resistant and persist until the end-stage of the disease, enabling paralyzed patients to communicate through ocular tracking devices. MNs of the Onuf’s nucleus in the sacral spinal cord, that innervate sphincter muscles and control urogenital functions, are also spared throughout the disease. There is also a differential vulnerability among MNs that are intermingled throughout the spinal cord, that directly relate to their physiological properties. Here, fast-twitch fatigable (FF) MNs, which innervate type IIb muscle fibers, are affected early, before onset of clinical symptoms, while slow-twitch (S) MNs, that innervate type I muscle fibers, remain longer throughout the disease progression. The resilience of particular MN subpopulations has been attributed to intrinsic determinants and multiple studies have demonstrated their unique gene regulation and protein content in health and in response to disease. Identified factors within resilient MNs have been utilized to protect more vulnerable cells. Selective vulnerability may also, in part, be driven by non-cell autonomous processes and the unique surroundings and constantly changing environment close to particular MN groups. In this article, we review in detail the cell intrinsic properties of resilient and vulnerable MN groups, as well as multiple additional cell types involved in disease initiation and progression and explain how these may contribute to the selective MN resilience and vulnerability in ALS.

## 1. Introduction

The primary targets of amyotrophic lateral sclerosis (ALS) are corticospinal motor neurons in the motor cortex (also called “upper MNs”), and somatic MNs in the brainstem and spinal cord (termed “lower MNs”). The lower MNs control voluntary contraction of striated skeletal muscles and thus their progressive breakdown and the loss of synapses between MNs and muscle (the neuromuscular junctions, NMJs) in ALS is responsible for the irreversible paralysis of the body (Figure 1). Lethal respiratory failure follows loss of phrenic nerve MNs and subsequent atrophy and weakness of the diaphragm and marks the end-point of the disease. Early dysfunction of brainstem (branchial) MNs, which control muscles derived from the pharyngeal arch, is responsible for the so-called bulbar onset of the disease, where the ability to make facial expressions, chew, swallow and speak are lost first, with the most severe prognosis to ALS patients. Visceral MNs that innervate cardiac and smooth musculature and form part of the autonomic nervous system do not generally succumb in ALS, although there are rare reports of autonomic dysfunction in advanced ALS reviewed in [1]. Why visceral MNs are resilient while somatic motor neurons degenerate is currently rather unclear. Furthermore, there is a differential vulnerability among somatic MNs which is apparent from different perspectives:

(1)Alpha MNs (αMNs), which innervate extrafusal muscle fibers, are vulnerable in contrast to gamma MNs (γMNs), which innervate the muscle spindles in the same muscle group [7].(2)Specific lower MN groups are more resistant than others. In particular, oculomotor, trochlear, and abducens MNs are spared in ALS until the end-stage of the disease [8,9] as well as in animal models of ALS, in contrast to other brain stem motor nuclei [10,11,12,13]. This permits eye-tracking devices as communication tools for fully paralyzed patients [14]. Another example are Onuf’s nucleus MNs in the sacral spinal cord, which control external sphincter muscles, which also remain throughout the disease [15,16,17].(3)There are gradients of vulnerability among somatic αMNs that innervate one particular muscle, correlating with MN and muscle activity and innervated fiber types. Specifically, fast-twitch fatigable (FF) alpha MNs are more sensitive to degeneration than slow (S) MNs, as shown in ALS patients [18] and ALS mouse models [19,20,21].

In order to understand these differences in vulnerability and resilience, we explored the motor system and its particular dysfunction in the course of the disease and discussed the possible contributions of several non-neuronal cell types, including interneurons, astrocytes, oligodendrocytes, microglia, Schwann cells, muscle, and peripheral macrophages.

## 2. Motor Neurons

### 2.1. The Primary Motor Circuit

The human motor system encompasses central nervous system structures in the cerebral cortex, brainstem, spinal cord, and cerebellum as well as nerves, skeletal muscles, and the specialized synapses between somatic MNs and muscles, the NMJs. Voluntary movement emerges from a cascade of neuronal activity across the components of the primary motor circuit; the planning and initiation of movement originates in the motor cortex by activation of the upper MNs, which are large pyramidal cells that are called Betz cells in primates. They project their axons through pyramidal tracts to lower MNs that execute motor functions. Here, the corticobulbar tracts connect to cranial nerve nuclei in the brainstem while the corticospinal tracts connect to MNs that are located in the ventral horn of the spinal cord (Figure 2). The lower MNs in the spinal cord and cranial motor nuclei project to the body’s periphery where they innervate and control skeletal muscles.

### 2.2. Divisional Motor Neuron Identity in the Spinal Cord

Lower MNs and their target muscle fibers form functional entities called *motor units*. Their functional specialization has greatly diversified during vertebrate evolution. In the human body, there are an estimated 100,000–450,000 spinal and brain stem MNs that innervate the approximately 100–200 million muscle fibers organized in more than 300 paired muscles [22]. Distinct groups of spinal MNs emerged successively during evolution and recapitulate evolutionary hallmarks such as the acquisition of limbs or of a diaphragm. These MNs localize in specific regions and form paired motor columns spanning the rostral caudal axis of the spinal cord [23]. The motor columns can be further subdivided into motor pools which constitute all the MNs that innervate an individual muscle and have been mapped in many vertebrates [24,25].

The ancestral medial motor column (MMC) is found in all vertebrates, spans the full spinal cord lengthwise, and innervates dorsal epaxial musculature (vertebrate extensor muscles). The ventral hypaxial musculature is innervated by the hypaxial motor column (HMC). At thoracic levels, the intercostal and thoraco-abdominal muscles belong to hypaxial musculature, and at postcrural level, prevertebral and the pelvic floor muscles. The diaphragm also arises from hypaxial muscles at the cervical level in the embryo but migrates caudally during development [26]. The phrenic motor nucleus has therefore been considered a part of the HMC by some authors [27] and by others as a specialized motor column [28]. In tetrapod animals, the lateral motor column (LMC) has emerged from the HMC at limb-innervating levels of the spinal cord with MNs that innervate the limb skeletal musculature. Likewise at the thoracic levels, the preganglionic column (PGC) replaces the LMC [25], which constitutes the visceral MNs of the spinal sympathetic and parasympathetic systems. The brachial motor column of the spinal cord at level C1-C3 is an extension of the cranial motor column of the brainstem with distinct origin from the LMC and is characterized by the expression of the transcription factor *Phox2b*, similarly to cranial MNs [29,30].

On the molecular level, the motor columns are distinguished by the expression of specific transcription factors during embryonic development [27,31,32] and by *hox* gene activation [33,34]. *Hox* genes regulate two aspects of the formation of the motor system. The colinear activation of sequential *hox* genes informs the position within the rostro-caudal body axis and columnar specification. Within each motor column, the *hox* gene expression further segregates it into individual motor pools [34] Transplantation studies in chick embryos suggest that MNs acquire their motor pool identity and their specificity for a particular target muscle before axon outgrowth occurs [35]. In fact, the coordinated activity of several *hox* genes orchestrate both the motor neuron pool specification and the connectivity of these motor neurons through regulating expression levels of *Ret* (encoding a receptor tyrosine kinase for members of the GDNF family) and the *Gfrα* co-receptor (which GDNF binds to) [36].

Motor axons from several MNs intermingle in the spinal nerves as they grow and consolidate into nervous plexi that are exposed to the same guidance signals. At the base of the limb, motor axons group into muscle-specific fascicles, a step that is regulated by PSA-NCAM and required for correct targeting of muscles during development [37,38,39] and also after injury and subsequent regeneration of motor axons [40]. Therefore, the early specification of motor pools and matching skeletal muscle by regulatory *hox* gene expression ensures faithful and proper connectivity of the motor units.

The columnar identities are further determined by interaction with downstream transcription factors. *Lhx3* is initially expressed in all MN progenitors and adjacent V2 interneuron progenitors [41,42]. *Lhx3* is not required for the generation of MN progenitors but for their diversification into motor columns, and it becomes restricted to the MMC in mice at later stages [41]. LMC and PGC MNs are characterized by the expression of *Foxp1* [43]. *Foxp1* is clearly pivotal to LMC identity as removal of this transcription factor causes loss of this motor column and the reversal to a more ancestral state where HMG MNs are generated, but not LMC or PGC MNs, followed by the expected alterations in axonal projection [43].

### 2.3. Alpha, Beta, and Gamma Motor Neurons

Somatic MNs are classified into alpha (α), gamma (γ), and beta (β) based on the innervation pattern in skeletal muscle fibers [22,44,45] (Figure 3). There are many proxies to assign MN subtypes including size morphology, the presence and absence of cholinergic synapses (C-boutons) on MN somas, and marker expression (see below). αMNs have large somas (80–120 µm diameter in humans) and large caliber axons (12–20 µm). They innervate the extrafusal fibers of skeletal muscles and are responsible for the main contractile force of skeletal muscles. Five percent of all direct projections from the motor cortex are to αMNs alone. αMNs are the most abundant MN subtype and are capable of high transduction velocities (80–120 m/s). γMNs constitute about 30% of all spinal MNs [46] and they innervate intrafusal fibers forming muscle spindles. γMNs do not cause any motor activity but control the muscle tonus by modulating the sensitivity of muscle spindles to stretch. They have smaller somas and axon calibers (5–8 µm) than αMNs and conduct signals slower (4–24 m/s). βMNs innervate both extra- and intrafusal fibers, where they serve as secondary receptors in muscle spindles [47,48]. They are intermediate in soma size, axon caliber (6–12 µm), and transduction velocities (33–75 m/s) compared with the other two subtypes and they make up about 10% of axons innervating the muscle spindles [49] (Figure 3). There are no molecular markers for βMNs, and their definite characterization remains electrophysiological and based on innervation pattern [50].

The differences between these three MN classes emerge late in development (at a postnatal stage in mice) [51,52,53,54] and few markers have been established to distinguish them immunologically in rodents. The transcription factor Esrrg (Err3) is restricted to γMNs two weeks after birth [55]. αMNs strongly express Rbfox3 (NeuN) [56], while γMNs lack this marker and express higher levels of the GDNF receptor *Gfra1* than the other MN subtypes [49,55,57]. γMNs also express very low levels of the transcription factor Mnx1 (Hb9) which is highly expressed in αMNs [49]. Immunohistochemical analysis in rats indicates that the enzyme Atp1a3 (alpha3 Na^+^,-K^+^-ATPase) is a selective marker of small axon fibers that innervate skeletal muscle stretch receptors, which presumably are γMN axons [58,59].

Transcriptomic characterization using single cell or single nuclei sequencing has corroborated the diversity of neurons in the developing and adult spinal cord [60,61,62]. Although somatic MNs are robustly identified histologically by a combination of cholinergic markers such as CHAT and SLC18A3 (commonly called VACHT) and their location in the ventral horn of the spinal cord, these studies have highlighted the challenge of characterizing spinal MNs in dissociated tissue. Due to technical limitations, the rarity of cholinergic neurons in the spinal cord, in combination with their exceptional size (including long processes), only a few cholinergic neurons were actually sequenced in these first studies. Therefore, cholinergic spinal neurons were enriched using genetic GFP markers in recent single nuclei sequencing studies that characterized many subpopulations of cholinergic interneurons, as well as visceral and somatic MNs in the mouse spinal cord [63,64]. Both studies identified *Tns1* as a novel and specific marker of somatic MNs in the spinal cord [63,64]. Its specific expression in somatic MNs was confirmed by in situ hybridization that co-stained large CHAT^+^ and PRPH^+^ neurons [65,66] in the ventral horn of the spinal cord [64]. Both Alkaslasi and colleagues and Blum and colleagues demonstrated that cholinergic neurons expressing *Zeb2* and *Fbn2* are visceral MNs [63,64], instead of αMNs, as previously proposed [62]. This is also in line with an earlier report that *Zeb2* demarks a visceral MN population [31]. Both studies identified three major subpopulations of somatic MNs, which may correspond to αMNs, βMNs, and γMNs [63,64]. However, assignment of βMN identity thus far relies on anatomical investigation and electrophysiology [50] as there are yet no established molecular markers for this MN group. αMNs are classically marked by high expression of Rbfox3 (also called NeuN), while γMNs are classified based on their lack of this marker reviewed in [22]. *Sv2b* and *Stk32a* were colocalized by in situ hybridization with *Rbfox3* in large cholinergic ventral horn neurons that had C-boutons on their somas and VGLUT1^+^ synapses, strongly supporting that these are new markers for αMNs [64]. γMNs segregated based on the previously identified markers *Esrrg* and *Gfra1* [55,57]. However, the expression pattern matched two highly similar subpopulations [63,64] that await further clarification. It is speculative at the moment whether these two populations represent γMNs and βMNs or maybe even different γMNs (dynamic and static), and further studies are needed to clarify their molecular identity. 

αMNs are vulnerable in ALS, while βMNs and γMNs appear resilient to this devastating disease. This has been demonstrated by careful quantifications in SOD1^G93A^, Prp::hTARDBP^A315T^, t^ON^::hFUS^P525L^ mouse models, which showed that γMN numbers remained unchanged at the end-stage of the disease when more than half of αMNs had been lost [7]. So why is that? Is this difference due to cell intrinsic properties of the MNs, cell extrinsic influences, or a combination of both? Regarding cell intrinsic properties that may regulate vulnerability in ALS, it is evident that αMNs with their larger axon calibers have a much greater cellular volume to support than the two other cell types and thus will require much more energy, which puts them at risk. When it comes to extrinsic properties that may help explain the selective resilience of γMNs, it has been shown that lack of excitatory input onto γMNs may protect these in ALS [7]. αMNs have input from primary afferent I_A_ sensory fibers while γMNs lack this, similar to oculomotor MNs, which are also resilient in ALS [67,68]. MNs of the Onuf’s nucleus, which control urogenital functions and are resilient in ALS [16,69,70], also appear to lack excitatory input as demonstrated by their lack of VGLUT1 inputs [7]. Thus, this is a common feature of resilient MN groups. Crossing of SOD1^G93A^ mice with Egr3 knock out mice, in which muscle spindles degenerate and muscle spindle afferents are eliminated, was protective to αMNs. Likewise, the dampening of afferent sensory signal by selective elimination of half the γMNs in the SOD1^G93A^ mouse, due to knock out of muscle spindle-derived GDNF on which γMNs depend, increased the survival of the mice [7]. This collectively supports the idea that reducing muscle afferent activity or the muscle’s sensitivity to stretch improves αMN survival and indicates that lack of I_A_ proprioceptive inputs onto γMNs protects them from ALS [7]. However, sensory afferent input from I_A_ proprioceptors is not generally required to elicit MN degeneration because some vulnerable αMN populations also lack this. For example, there is an early disruption of NMJs of MNs that innervate the cutaneous maximus muscle (a type II muscle) (by P60 already) in the SOD1^G93A^ mouse [71], although these MN somas lack monosynaptic sensory afferent proprioceptive input altogether [72]. Likewise, only a few hypoglossal MNs actually have VGLUT1^+^ synapses [73], although they are vulnerable in ALS. Altogether, I_A_ sensory input could contribute to the difference between αMN and γMN sensitivity in ALS as it may exacerbate hyperexcitability of αMNs, but it does not appear to underlie the vulnerability of αMNs fundamentally.

### 2.4. Fast and Slow Alpha Motor Neurons and Their Differential Vulnerabilities

The identity of αMNs in the motor unit is even further informed by the contractile properties of the skeletal muscle fibers they innervate; αMNs controlling either fast-twitch fatigable (FF) muscle fibers, or fast-twitch fatigue-resistant (FR) muscle fibers, or slow-twitch (S) muscle fibers form separate subtypes [74]. Transduction burden further correlates with the neuron size among αMNs, with FF being larger on average than S MNs (size principle) [75,76]. There is a large variability in soma sizes among the subpopulations of αMNs, but on average S, FR, and FF soma sizes do not seem significantly different [77,78,79,80]. However, the main volume of an MN is in its axon (the axon can be 20,000 times longer than the soma diameter), which constitutes >98% of the volume of a spinal MN. As FF MNs have more than twice the axon diameter than S MNs (12–20 μm for FF compared to 5–8 μm for S) and more extensive branching (FF MNs innervate 300–2000 muscle fibers while S MNs innervate less than 200 fibers), they by far exceed S MNs in total volume, reviewed in [81]. In addition, FF MNs have more extensive dendritic branching than S MNs [82].

Among spinal αMNs, there is also a gradient of vulnerability. The NMJs between FF MNs and muscle are denervated first before NMJs of FR and S MNs are affected. S MNs show the highest resistance among these spinal MN groups [20,83,84] (Figure 4). FF MNs appear particularly prone to ER stress and show a gradual upregulation of ER stress markers shortly after birth in mutant SOD1 mouse models of ALS, followed by unfolded protein response, microglial activation and axonal degeneration. Salubrinal, a compound which dampens ER stress, could delay disease progression [85].

S MNs can partially compensate for the denervation of NMJs on FF muscles by axon sprouting and reinnervation of vacated endplates before they themselves also succumb to the disease and retract their axons [19,20,86]. As S MNs innervate vacated motor endplates on muscle fibers previously innervated by FF MNs, they switch the type of muscle fiber from fast to slow (Figure 1 and Figure 4) [2,87,88]. If we could harness the regenerative capacity of S MNs and stabilize new NMJs formed during disease progression, we could perhaps slow down disease progression. It has been suggested that it is the low level of EPHA4 in S MNs, compared to FF MNs, that render them more capable of regeneration in ALS. Modulating *Epha4* signaling successfully prolonged the life span of ALS mice [89]. Axon sprouting, and reinnervation of vacated endplates can also be induced by the delivery of growth factors, such as IGF2 in ALS [90]. Combining treatments targeting both ER stress and axon regeneration by blocking *Eph* signaling and stimulating growth factor signaling may be the way forward.

### 2.5. Molecular Underpinnings of Oculomotor Neuron Resilience to Degeneration in MN Diseases

Not all somatic lower MNs in the brain stem and spinal cord are vulnerable to degeneration in ALS. MNs belonging to three cranial nerves (III, IV, and VI), oculomotor, trochlear, and abducens, are particularly resilient to degeneration in ALS [8,9,11,12], reviewed in [81]. Multiple labs including ours have globally analyzed the intrinsic properties of oculomotor and trochlear MNs in vivo and in vitro to understand their resilience and have shown that these MNs have a unique transcriptional code that sets them apart from other, more vulnerable MN groups [11,91,92,93]. 

Part of the signature of CNIII/IV MNs that we believe renders them resilient in ALS are their relatively high intrinsic levels of IGF1 and IGF2 [11,94]. CNIII/IV MNs also show higher levels of phosphorylated IGF receptors (IGF-Rs) on their surface, while none of the surrounding cells appear to express IGFs, thus indicating the potential for a cell autonomous effect where IGFs are secreted from CNIII/IV MNs, bind to IGF-Rs on their own surface in an autocrine fashion and consequently stimulate *Akt* signaling and promote survival [90]. Delivery of IGF2 to vulnerable MNs in SOD1^G93A^ mice using adeno-associated viral (AAV) delivery protected MNs and their synapses with muscle and caused a regenerative response in motor axons and prolonged the life span of the mice by 10% [90]. Oculomotor MNs (OMNs) also harbor intrinsically high levels of Synaptotagmin 13 (SYT13) across mice, rats, and humans [95]. SYT13 protected human spinal MNs from degeneration in vitro and appeared to block ER stress and apoptosis. In vivo gene therapy with SYT13, using AAV, improved the life span of SOD1^G93A^ mice by 14%. Remarkably, SYT13 also improved the survival of SMA mice by 40%. The specific function of SYT13 in MN protection across MN diseases is still under investigation, but it is evident that SYT13 protects both MN somas and axons from degeneration across ALS causations [95].

Several studies have indicated that OMNs may have a reduced susceptibility to excitotoxicity which may be mediated partly through enhanced GABAergic transmission and high calcium buffering capacity [11,90,92,94]. CNIII/IV MNs are, of course, residing in a milieu that is different from, for example, spinal MNs. It is not yet entirely clear how much the environment surrounding CNIII, IV, and VI MNs contributes to their selective survival in ALS. What is known is that OMNs are relatively resilient to degeneration; also, if generated from stem cells and cultured in ALS-like conditions in vitro, induced by excitotoxicity through kainic acid [93].

OMNs may not only be resistant by expressing a number of neuroprotective factors, but perhaps also by lacking a combination of degeneration-inducing molecules. Kaplan and colleagues showed that vulnerable FF MNs have higher levels of the metalloprotease MMP9 than OMNs and Onuf’s neurons [91]. In this elegant study, it was demonstrated that MMP9 overexpression could accelerate the degeneration of FF MNs in the SOD1^G93A^ mouse, while suppression of MMP9 delayed denervation. However, overexpression of MMP9 in resistant OMNs or slow MNs innervating the soleus muscle in the SOD1^G93A^ mouse was insufficient to induce degeneration of these neurons [91]. Thus, it is evident that MMP9 alone does not explain vulnerability or resilience but may be part of the puzzle.

Are resilient neurons intrinsically similar in some way that explains their resilience in ALS or do they respond to disease in a way that clearly counteracts induced toxicity or both? To investigate this matter, we conducted RNA sequencing of MNs isolated from the oculomotor nucleus and the Onuf’s nucleus in human postmortem tissues. We then compared the genes that were differentially expressed in these resistant neurons compared to vulnerable spinal MNs. We found that the vast majority of differentially expressed genes were unique to each resistant motor nucleus, which indicates a resilience-signature that is unique to each nucleus. Nonetheless, a number of transcripts were common to both resilient MN groups and among these were *MIF* (macrophage migration inhibitory factor) [93] which has been shown to be neuroprotective in SOD1-ALS [96].

It remains to be studied how oculomotor MNs respond dynamically to mutations that cause ALS and if such a response is unique and further explains their resilient nature. From neuropathological studies of TDP43 mislocalization and aggregation in ALS, we do know that oculomotor MNs show signs of pathology, while still remaining apparently intact.

In fact, phosphorylated-TDP43 (pTDP43) inclusions can be seen in OMNs in ALS (50% of cases in one study) but also in normal aging [97]. This could indicate that the normal aging process is a contributor to accumulation of pTDP43 aggregates in the cytoplasm. However, it should be noted that the fine structure and level of pTDP43 with aging appeared different than that of ALS, with more thin filamentous inclusions and lack of skein-like aggregations [97]. This is perhaps an indication that normal aging induces a less aggressive level of protein mislocalization than ALS. Can inclusions be correlated with disease severity and thus demonstrate how poorly a neuron is doing? It appears that this may be the case, as it was recently shown that severity of pTDP43 pathology and neuronal loss correlated and that only in very late stages of diseases were pTDP43 inclusions in OMNs apparent. Notably, at this very late disease stage (stage 5), midbrain dopamine neurons were almost completely lost indicating that by the time OMNs start to be affected, ALS has expanded into multiple other non-motor systems that appear more vulnerable to ALS than OMNs. Non-cell autonomous aggregation of pTDP43 in oligodendrocytes in both gray and white matter appeared to correlate with the neuronal loss [98]. Spiller and colleagues set out to investigate if TDP43 mislocalization and aggregation triggers death in all cells or if this is cell-type specific. They used a doxycycline-suppressible transgene engineered to express human TDP43 with a defective nuclear localization signal under the control of the human neurofilament heavy chain promoter. They found widespread TDP43 pathology across MN subpopulations after 8 weeks of inducible expression, but while MNs within the hypoglossal nucleus and spinal cord started to degenerate, oculomotor, trigeminal, and facial nuclei were spared [13]. This demonstrates that while TDP43 mislocalization and accumulation is linked to MN degeneration, certain MNs appear to cope with such protein aggregates better. If we were to study the dynamic response of differentially vulnerable MNs to TDP43 accumulation, we would likely gain mechanistic insight into this process.

We have so far analyzed the transcriptomic profiles of human OMNs in control and ALS at disease end point, which gave some important clues. Firstly, we found that *SYT13* levels remained high at the end-stage of the disease in both OMNs and in remaining relatively resilient spinal MNs [95]. It remains to be investigated if *SYT13* was initially high in the spinal MNs that remained until end-stage, and presumably were of an S MN type, or if *SYT13* was upregulated in a protective response to disease. Future single-cell analysis in tissues and in iPSC-derived MNs could resolve this matter. Secondly, we also identified relatively high levels of *NDNF*, (neuron-derived neurotrophic factor), *NRSN1* (Neurensin), and *PLPPR4* (phospholipid phosphatase-related protein type 4) across relatively resilient neurons at the end-stage of the disease compared to control, which may aid in neuroprotective responses [95]. Towards the goal of fully comprehending how OMNs remain intact in ALS, it would be very interesting to study the dynamics of the transcriptome in OMNs in response to sporadic disease and also familial ALS-causative mutations either in animal tissues or stem cell-derived neurons. We have conducted such a study in SMA (spinal muscular atrophy) mice, which showed that each neuron population responded in a unique and temporally regulated fashion so that very little overlap in differential gene regulation was seen across disease stages. Thirdly, we found that OMNs elicited a neuroprotective response, that seemed to counteract detrimental processes indicating DNA damage and steps towards apoptosis, inducing genes such as *Gdf15*, *Lif*, *Chl1*, *Cplx2,* and *Aldh4a1*, as well as regulation of motor proteins *Kif5a* and *Kif3a*, which could aid in axon transport [99]. Interestingly, *KIF5A* is associated with ALS [100,101] and adult-onset distal spinal muscular atrophy [102], highlighting the importance of axonal transport for MN function. Both *GDF15* and *LIF* are trophic factors that appear to have important roles in maintaining MN survival and connectivity with muscle in vivo [103,104,105,106]. We used GDF15 to promote human spinal MN survival in vitro [99]. This further solidifies previous findings that resilient OMNs contain high levels of certain growth factors with known MN survival properties, including IGF1 and IGF2, and with activation of downstream survival signaling pathways such as PI3/AKT [11,90,93,94]. Finally, it is evident that activation of pathways preferential to resilient oculomotor MNs can induce resilience in vulnerable MNs and, as such, oculomotor MNs are viable therapeutic screening tools.

## 3. Interneurons

### 3.1. Cortical Interneurons

Interneurons form essential regulatory circuits in the central nervous system and, as such, modulate both upper and lower MN activity. ALS patients display cortical hyperexcitability that is thought to arise from an imbalance between excitatory cortical MNs and inhibitory cortical interneurons. The underlying mechanisms are not resolved, but ALS patients appear to have abnormal expression patterns of neurotransmitter receptors in the motor cortex, e.g., reduced level of GABA_A_ receptors [107] and a generally lower efficacy of inhibitory inputs in the cortex [108]. Further, the abnormal intracortical excitability can be normalized by drugs improving GABAergic transmission [109,110]. Evidence of increased excitatory activity (network drive) also coincide with the onset of tremors in the SOD1^G93A^ mouse model [111,112,113]. These reports indicate that changes in the connectivity between MNs and interneurons contribute to the cortical ALS pathophysiology.

### 3.2. Spinal Interneurons

The selective vulnerability of different lower MN subtypes in ALS correlates with their function and intrinsic properties. However, they also integrate specific regulatory inputs from interneurons which, therefore, also correlate with the patterns of vulnerability in MN diseases [114]. Some loss of spinal interneuron has been observed in postmortem ALS spinal cord [115,116] and in several SOD1 mouse models, but only at late stages after large numbers of MNs have already succumbed [117,118]. Still, the early changes observed in synaptic inputs from interneurons to MNs could contribute to vulnerability or resistance [117].

Lower MNs in the brainstem and spinal cord express high levels of glycine and GABA_A_ receptors that modulate their excitability and firing rates [119]. Glycine and GABA are the neurotransmitters used by several types of spinal interneurons: V0_D_ and V0_S_ interneurons, V1 interneurons including Ia interneurons and Renshaw cells, as well as V2b interneurons [120,121]. The pattern of inhibitory receptors on ALS-resistant spinal MNs is different from vulnerable MNs to the effect that it apparently strengthens glycinergic over GABAergic neurotransmission [122], likely representing different synaptic input on vulnerable and resilient spinal motor neuron populations. Overall, the glycinergic receptors are reduced in the spinal cord of ALS patients [123]. Moreover, several other studies have reported a reduction of glycinergic but not GABAergic input in SOD1^G93A^ mouse models starting as early as from day 60 before MNs are lost [124,125,126,127].

V1 Renshaw cells are recurrently inhibitory interneurons on MNs that receive their input from motor axon collaterals in the spinal cord. They inhibit the same MN that activates them alongside proximal MNs and reduce their firing, allowing for higher pulse rates [128]. Notably, recurrent inhibition is decreased in ALS patients, but not in patients with spinal cord injuries [129]. Renshaw cells marked by calbindin are not depleted before MNs are lost in SOD1^G93A^ mice [117,130]. However, there are early changes that indicate a decoupling of the Renshaw cells from the cholinergic input of the motor axon collaterals with downregulation of the acetylcholine receptor *Chrna2* in Renshaw cells and *Slc18a3* (Vacht) on the MN side [117]. At the other side of the circuit, the synapses from Renshaw cells onto MNs undergo complex changes in SOD1^G93A^ mice that indicate pruning, possibly because of reinnervation attempts to remaining MN somas [117]. It is noteworthy that the connectivity between Renshaw cells and MN subtypes is not equal and fast MNs form more synapses with Renshaw cells than slow MNs [131]. Therefore, a dysfunction in the recurrent inhibitory circuit could affect FF MNs more than resilient S MNs.

Allodi and colleagues uncovered MN subtype-specific alterations in the interneuron MN transmission in the spinal cord of SOD1^G93A^ mice [124]. They found that FF MNs received stronger glycinergic input than S MNs and that glycinergic synapses on FF MNs are selectively lost in the SOD1^G93A^ disease progression. The loss of spinal glycinergic inputs on αMNs resulted in specific locomotor defects, including slowed down locomotion and reduced stride length. This locomotor phenotype agrees with the selective loss of V1 interneurons (marked by EN1) in ALS mice and is phenocopied in wild-type mice with reversible silencing of spinal EN1^+^ neurons, specifically [124]. Similar locomotion defects were observed in ALS patients [132].

V1 interneurons form the majority of glycinergic interneurons in the spinal cord (ca. 80%) and they contribute substantial input to αMNs with 50% of near soma MN synapses. While the strong reduction of the V1 glycinergic inhibitory inputs to MNs could improve excitability at early stages [127], the lower inhibitory control during disease progression could result in the observed MN hyperexcitability phenotype and may put a higher excitotoxic burden on vulnerable FF MNs.

## 4. Astrocytes

Astrocytes are non-myelinating glial cells in the central nervous system that have a multitude of functions and an astounding number of processes that envelop neuronal synapses. In humans, a single astrocyte can interact with up to 2 million synapses at a time [133]. Many astrocytes express GFAP (glial fibrillary acidic protein) which is an intermediate filament. The main function of GFAP is to maintain astrocyte structural integrity and aid in cell movement and change of shape [134]. When astrocytes become activated, for example, during inflammatory processes seen in neurodegenerative disease such as ALS (Figure 1), they show an increased expression of GFAP (Figure 5). Astrocytes are generally divided into two main categories, fibrous (in the white matter) and protoplasmic (in the gray matter). Fibrous astrocytes have relatively few organelles, exhibit long unbranched processes, and often have end-foot processes which connects them to the outside of capillary walls when in proximity to these. Protoplasmic glial cells are most prevalent and found in the gray matter and these possess a larger quantity of organelles and have highly branched processes (Figure 5a). These astrocytes extend end feet to blood vessels and are, here, part of the blood–brain barrier. There are also radial glial cells which have processes that go from the pia mater to the gray matter. These are mainly known to have a functional role in development when glial and neurons migrate along these processes [135].

Astrocytes promote neuronal survival, induce functional synapse formation (irrespective of the age of the host the astrocytes originate from), and engulf synaptosomes. Astrocytes are heavily involved in spatially regulating extracellular potassium concentrations through the expression of a number of potassium ion channels, KIRs (potassium inward rectifying channels). Repolarization of neurons tend to raise potassium concentrations in the extracellular fluid (which is normally low in potassium ions and high in sodium ions). If the rise is significant, it can interfere with neuronal signaling by depolarizing neurons. Astrocytes remove potassium ions from the extracellular space and distribute the ions throughout their cytoplasm and further to their neighbors through gap junctions, keeping potassium at levels extracellularly that do not interfere with normal propagation of an action potential [137]. Astrocytes also provide trophic support to neurons by secreting a number of different growth factors including BDNF, GDNF, and NGF. However, the levels of trophic factors seem to decline with age in certain regions of the CNS and appear to be lower in humans than in mice [138]. It has been shown that multiple cell types show an accelerated aging phenotype in neurodegenerative diseases such as ALS, with increased DNA damage as a result. While it has not yet been directly demonstrated that astrocytes show DNA damage in ALS, there is indirect evidence thereof reviewed in [139]. Consequently, if astrocytes then age unreasonably fast in ALS, this could result in a lower secretion of trophic factors to surrounding neurons, thus leaving these less supported, but this remains to be further investigated.

Astrocytes are clearly affected by ALS in multiple ways and take an active part in the disease. In transgenic SOD1 mice, astrocytes show early Lewy-body-like inclusions, immunoreactive for SOD1, prior to glial activation and prior to the accumulation of inclusions in neurons [140].

Mislocalization of TDP43 is a common neuropathological finding across ALS causations, with implications for both toxic gain of function as well as loss of function. In addition to the mislocalization of TDP43 in neurons, astrocytes in ALS patient tissues show cytoplasmic TDP43 inclusions [141].

As evidenced from animal studies where mutant SOD1 was removed specifically from astrocytes in the SOD1^G37R^ ALS mouse (using a GFAP-Cre mouse), astrocytes appear to drive late disease progression. Here, disease onset was unaffected but microglial activation was reduced (fewer MAC2^+^ iNOS^+^ microglia) and concomitantly with that, later disease progression was slowed down [142]. While astrocytes are mainly implicated in disease progression, it was shown that a high level of overexpression of mutant TDP43 in astrocytes alone is sufficient to trigger MN death in animals [143].

Astrocytes also play a critical role in regulating extracellular neurotransmitter levels, particularly important for excess glutamate which is sequestered through excitatory amino acid transporter 2 (EAAT2 or GLT1) on astrocytes. Loss of rat GLAST (EAAT1) or EAAT2 after injection of antisense oligos into lateral ventricles resulted in elevated extracellular glutamate levels and induced signs of neuropathology, including cytoplasmic vacuolization in neurons in the striatum and hippocampus [144]. ALS patients have been described to show loss of EAAT2 in the motor cortex and spinal cord [145]. Transgenic SOD1^G85R^ ALS mice phenocopy this feature of the human disease and show a clear downregulation of EAAT2 in spinal cord [140]. Similarly, transgenic SOD1^G93A^ rats show a downregulation of EEAT2 in the ventral horn at symptomatic stages and a complete absence at end-stage [146]. Thus, the loss of EAAT2 on astrocytes would leave MNs exposed to increased synaptic glutamate levels, which could lead to excitotoxicity. So how would the loss of EAAT2 tie-in to selective vulnerability? It is possible that EAAT2 levels remain high on particular subpopulations of astrocytes and thus lead to low glutamate exposure of MNs in particular locations. As S and FF MNs have input from distinct astrocyte populations (as discussed above), these may show differences in EAAT2 regulation and thus subsequent differences in protection of surrounding MNs. Spatial transcriptomics has been used to look at glial activation in ALS [147], but as the resolution at that time was about 100 µm, it was not possible to resolve reactivity and gene expression differences at the single-cell level. However, this matter could be resolved with single-cell RNA-FISH where a combination of probes against EAAT2, GFAP, KIR4.1, and specific markers for S and FF MNs could reveal if there is heterogeneity in astroglial as well as neuronal responses to ALS.

Regarding regional specificity, it was early on shown that the shape and branching of midbrain dopamine (mDA) neurons can be influenced by coculturing these with astrocytes originating from different regions of the brain. When mDA neurons were grown on a monolayer of astrocytes from the striatum, they showed minimal extension of processes. However, when the mDA neurons were instead cultured on astrocytes derived from the mesencephalon, they extended a large number of highly branched processes [148,149].

Astrocytes in the spinal cord show several levels of diversity with potential implications for neuronal function, e.g., they appear to have distinct dorsoventral positional identities, as shown in mice. In fact, there are three distinct populations of astrocytes that arise from the p1, p2, and p3 progenitor domains during early development. These subtypes of astrocytes can be identified on the basis of their expression of different transcription factors PAX6, NKX6.1, and the cell surface markers, reelin and SLIT1. Specifically, dorsally located VA1 astrocytes, derived from the p1 domain, express PAX6 and reelin; ventrally located VA3 astrocytes, derived from p3, express NKX6.1 and SLIT1; and intermediate white-matter located VA2 astrocytes, derived from the p2 domain, express PAX6, NKX6.1, reelin, and SLIT1 [136] (Figure 5b). Importantly, in vitro specification of human pluripotent stem cells into astrocytes shows that these can be patterned into distinct rostro-caudal positions as well as dorsoventral identities with distinct homeodomain transcription factor protein profiles [150]. So, do astrocytes diversify in a manner similar to the neurons they surround during development as would be expected if they are patterned by the same morphogen signaling centers (even if the timing of generation is lagging a bit and may affect levels of morphogens seen as many signaling centers are transient and show fate switches)? Then, if so, are their identities intrinsically linked and important to understanding vulnerability and resilience in neurodegenerative diseases such as ALS? The short and enticing answer seems to be yes for both questions. It was recently demonstrated that neurons and glial cells show shared region-specificity [151]. Thus, as neurons are diversified during development, so also are the glial cells surrounding them. So, do fast and slow MNs have input from astrocytes with distinct identities, and could this explain some of the differences in vulnerability between these MNs subtypes that is then in part cell-extrinsically regulated? This does indeed seem to be the case. It was recently shown that astrocytes in the spinal cord and particularly in the ventral horn of both mouse and man express high levels of KIR4.1, an inward-rectifying K^+^ channel, compared to astrocytes in outer CNS regions, including the dorsal horn of the spinal cord. Astrocytes abutting MMP9^+^ MNs, marking FF MNs, showed a higher level of KIR4.1 than those surrounding MMP9-negative MNs (Figure 5b). KIR4.1 expression appeared VGLUT1-dependent as knockout mice showed marked decrease in KIR4.1. Removal of KIR4.1 from astrocytes did not affect the number of MNs generated or surviving in the spinal cord, but affected soma sizes postnatally, where the largest soma sizes were missing. Labeling of MNs with an MMP9 antibody and through retrograde labeling from the tibialis anterior (TA) muscle, which mainly is composed of fast-twitch muscle fibers, confirmed the reduced αMN size in animals with KIR4.1 knockout in astrocytes. The animals also displayed a shift in electrophysiological properties to a more slow-like MN phenotype and had an accompanying decrease in muscle mass and grip strength, suggesting that FF MNs are selectively dependent on KIR4.1^+^ astrocytes to maintain their normal properties [152] (Figure 5c). Consistently, overexpression of KIR4.1 specifically in astrocytes was sufficient to increase the size of both FF and S MNs. Treatment with Rapamycin prevented this size increase, demonstrating a regulated through mTOR, a known modulator of neuronal size. It has previously been shown that KIR4.1 expression is progressively decreased in spinal cord astrocytes in SOD1^G93A^ mice, which may result in increased extracellular K^+^ which could lead to MN death [153]. Furthermore, ALS patient astrocytes, generated from iPSCs with a SOD1^D90A^ mutation, also showed decreased expression of KIR4.1. However, loss of KIR4.1 in astrocytes in the SOD1^G93A^ mouse did not further exacerbate MN death in this model [152] (Figure 5c). Collectively, these findings show that astrocyte KIR4.1 is essential for maintenance of peak strength and suggest that KIR4.1 downregulation might uncouple symptoms of muscle weakness from MN cell death in diseases like ALS.

What is the contribution of astrocytes to ALS? Do they just become less supportive, or do they actually become toxic and induce death? What is the evidence? Nagai and colleagues demonstrated that primary embryonic spinal MNs isolated from mutant SOD1^G37R^ mice had a mild cell-autonomous phenotype with smaller cell body diameters and shorter axon lengths compared to non-transgenic and wild-type SOD1-overexpressing MNs, and that mESC-derived MNs overexpressing mutant SOD1^G93A^ had a similar phenotype [154]. Healthy MNs had decreased survival when plated on transgenic SOD1^G93A^, SOD1^G37R^, or SOD1^G85R^ primary astrocytes compared to when plated onto normal astrocytes or wild-type SOD1-overexpressing astrocytes. They further demonstrated that toxicity to MNs was triggered by a soluble factor or factors from astrocytes, using conditioned media. This toxicity was specific to astrocytes as neither primary skeletal myocytes, cerebral cortical neurons, nor skin fibroblasts expressing comparable levels of mutant SOD1 caused reductions in MN numbers. Spinal cord microglia expressing mutant SOD1 also elicited toxicity to primary MNs, but it was milder than what was seen with astrocytes [154]. The mSOD1 astrocyte-toxicity appeared specific to MNs as spinal interneurons and dorsal root ganglia were left unharmed after 7 days of culture. It is compelling that toxicity is non-cell autonomous but still has an apparent selectivity for MNs. The astrocyte-mediated toxicity to MNs appeared to elicit a *Bax*-dependent cell death pathway, which could be alleviated by incubating cultures with the pentapeptide VPMLK, which inhibits BAX [154]. 

Simultaneously, di Giorgio and colleagues showed that mESC-derived MNs died to a higher proportion when cocultured on mutant SOD1-overexpressing astrocytes than when cultured on normal of wild-type SOD1-overexpressing astrocytes [155]. 

The following year, two laboratories showed that human MNs derived from embryonic stem cells were selectively destroyed by astrocytes overexpressing human mutated SOD1, confirming the studies on mice [156,157]. Here, factors responsible for the toxicity included prostaglandins [157] and proinflammatory cytokines [156]. Both studies concluded that the toxicity was mediated by secreted factors. Prostaglandin D2 (PDG2), one of the secreted factors identified, could by itself cause MN degeneration at a level similar to coculture with toxic astrocytes [157]. However, blocking the PDG2 receptors only modestly rescued MNs grown of mSOD1-astrocytes, indicating that mutant astrocyte-induced degeneration is mediated only in part by secreted PDG2; see commentary by [158]. Mutant SOD1-expressing astrocytes also secrete IFN-γ, which can induce degeneration of MNs in vitro [159]. Astrocytes derived from both sporadic and familial ALS patients are toxic to MNs and it may be that some processes causing this toxicity are common across causations, as it was shown that knocking down SOD1 in sporadic ALS improved MN survival [160]. However, the rescue by knockdown of SOD1 was variable in sporadic ALS lines and in many cases much less prominent than in familial ALS caused by mutant SOD1, and thus it is likely that multiple pathways are at play and that personalized medicine will be vital in modulating astrocyte toxicity.

There is no doubt that astrocytes become toxic and activated in ALS, but it remains to be investigated if particular MNs are selectively resilient to this toxicity and if so, why or if astrocytes differ in their activation and toxicity, and if this could explain selective MN vulnerability. Or, alternatively, if different MN populations are selectively resilient as they lack the receptors needed to be affected by detrimental signaling from astrocytes. Longitudinal spatial RNA sequencing of astrocytes and MNs in situ in ALS mouse models from early presymptomatic stages through the end-stage could determine this issue in an elegant manner.

Researchers have identified astrocytes with an aberrant phenotype in SOD1^G93A^ rats at symptomatic stages that had marked proliferative capacity and which appeared to lack replicative senescence when cultured in vitro (and could be propagated for 1 year), and termed these AbA cells (aberrant astrocytes). These AbA cells had relatively high levels of S100B and connexin 43 (CX43) while lacking GLT1, compared to control embryonic astrocytes (normal adult astrocytes could not be cultured to the same extent so comparisons were made with embryonic control astrocytes). Only 10% of primary MNs survived 48 h coculture with adult AbA cells from the symptomatic SOD1^G93A^ rat, while 60% of MNs survived coculture with neonatal astrocytes from the SOD1^G93A^ rat, as compared to 100% of MNs cultured on control neonatal astrocytes. Conditioned media from AbA cells gave the same result in MN death, and dilution of the media resulted in greater MN survival, indicating that AbA cells secreted factors that are directly toxic to MNs rather than being less supportive to MNs. The toxic effect of the conditioned media appeared selective to MNs as hippocampal neurons were not affected [161]. This study was particularly interesting in the terms of selective vulnerability, or rather lack thereof. In most studies, 50% of cultured MNs are killed by astrocytes, but here, 90% of the cells were rapidly lost. We do not know the composition of the MN cultures used and if they indeed contained both S and FF MNs, but we think it is safe to presume that S MNs would have been part of the culture and that these were killed off by the AbA cells together with FF MNs. It would, thus, be very interesting to identify soluble factors that are secreted from AbA cells and how these differ from regular SOD1^G93A^ astrocytes in an effort to better understand MN toxicity, selectivity, and resilience and vulnerability.

## 5. Microglia

Microglia are the resident macrophages of the central nervous system (CNS) and the most abundant glial cell of the CNS, making up 10–15% of all cells found in the brain. Microglia originate from yolk sac primitive macrophages and populate the CNS during early embryonic development and persist throughout adulthood. This early colonization of the embryonic CNS by microglia is conserved across vertebrate species, which implies that it is essential for early brain development. It has indeed been shown that microglia are pertinent for the wiring of circuitry in the forebrain [162], and in the adult brain, microglia are required for remodeling neural circuits, as well as modifying and eliminating synapses [163,164]. Microglia act both in immune defense and in maintenance of CNS homeostasis. Microglia are part of the innate immune system and thus constantly scan the environment for immediate danger from invading pathogens as well as damaged and dying cells. Through expression of toll-like receptors (TLRs), they can be become activated by molecules expressed by various pathogens such as lipopolysaccharide (LPS). They can also respond to ATP and HMGB1 released from damaged immune cells (two forms of DAMPs: damage-associated molecular patterns) through purinergic receptors (P2R) or TLRs or RAGE. Microglia respond to inflammatory molecules, e.g., TNF and CSF1, released from activated astrocytes and to aggregated proteins that accumulate in the CNS in neurodegenerative diseases, e.g., mutant SOD1 in ALS, beta-amyloid in Alzheimer’s disease, and alpha-synuclein in Parkinson’s disease. In response to any such stress signals, microglia initiate programs to try to resolve the problem and protect the CNS from harm and support tissue repair and remodeling. However, as microglia transition from a resting state to an activated state, they produce reactive oxygen species through NOX2 and NO through inducible nitric oxide synthase (iNOS), which are highly reactive and can damage molecules in the cell, including proteins, lipids, and DNA.

Microglia are rather uniformly distributed in the brain and spinal cord but show increased densities in neuronal nuclei, including, for example, the substantia nigra pars compacta. It was recently shown that there is significant heterogeneity in microglia across brain regions. Specifically, transcriptional profiling of microglia isolated from the cerebellum, cortex, hippocampus, and striatum of the adult brain revealed their heterogeneity and networks that set them apart were energy metabolism, immune response and regulation, oxidative stress, cell death/apoptosis, and endocytosis/vesicle transport [165]. The profiling suggests that microglia in different brain regions show a difference in vigilance to immune challenges with cerebellar and hippocampal microglia seeming more alert than, for example, cortical microglia. Functional testing of this hypothesis by challenging freshly isolated microglia with *E. coli* showed that cerebellar microglia were able to better control the net replication of bacteria than cortical microglia [165].

Microglia become activated early on in ALS, at or before disease onset in mutant SOD1 mice and with the number of cells increasing with progression [166,167] (Figure 1). Activated microglia first appear focused around ventral aspects of the spinal gray matter in mutant SOD1 mice, but then spread throughout the entire gray and white matter [166,168]. Using a reversibly inducible TDP43 in neurons, Spiller and colleagues could show that activated microglia selectively clear TDP43 aggregates and remain relatively suppressed in the ALS phenotype compared to control. Upon ending TDP43 expression, more microglia proliferated markedly and mice with early inhibition of microgliosis failed to regain full motor function [169]. These findings indicate that microglia are neuroprotective in early stages of ALS. These microglia found activated in the CNS of ALS mice do not derive from migrating macrophages to any great extent, although there is some influx through the blood–brain barrier during later stages of the disease [167,170]. SOD1^G93A^ ALS mouse microglia have a defined profile that is distinct from that of other types of activated microglia, for example, after LPS stimulation, and show upregulation of both toxic factors and potentially neurotrophic factors, e.g., IGF1 [167]. So far it has not been closely investigated if there is a regional divergence in activation in ALS that is cell intrinsically regulated rather than elicited in response to damage signals by the surrounding MNs.

When mutant SOD1^G37R^ was removed from microglia and macrophages in transgenic ALS mice, through crossing with a *Cd11b*-Cre mouse line, ALS disease progression was delayed significantly, while onset was unchanged, thus demonstrating that these cells are important for disease progression [168]. Similarly, repopulating the SOD1^G93A^ mouse model with wild-type myeloid progenitors delayed disease progression [171].

It has been suggested that microglia induce MN death via the NF-κB pathway, which is highly activated in microglia already from disease onset in the SOD1^G93A^ mouse [172]. Coculture of control Hb9::GFP MNs with adult SOD1^G93A^ microglia (but not neonatal) isolated from either brains or spinal cords of transgenic mice induced degeneration of 50% of MNs within three days. Reduction of mSOD1^G93A^ in adult microglia with 75% through shRNA left MNs unharmed, showing that the toxicity of the microglia was related to the prolonged overexpression of mutant SOD1. Suppression of NF-κB signaling in the SOD1^G93A^ microglia normalized the production of TNFα and nitric oxide (NO) and prevented microglia-induced MN death and axonal damage in vitro [172].

Activation of NF-κB in wild-type microglia caused 50% of the Hb9::GFP^+^ MNs in vitro to degenerate within 72 h of coculture, thus indicating that mutant SOD1^G93A^ microglia become toxic due to NF-κB activation. Knockdown of NF-κB signaling in microglia in vivo by Cre-mediated heterozygous removal of IKKβ, using the CSF1R promoter (colony stimulating factor receptor 1, that in the adult is only expressed in microglia) did not affect onset of disease but extended the disease progression phase by 47%. When animals were analyzed at the end-stage, there was a similar level of gliosis in the CNS of all mice, but if animals were age-matched and CSF1R-Cre mice were thus analyzed at the time of the Cre^–^ littermate end point, astrogliosis and microgliosis was decreased by 30% and 25%, respectively. This indicates that NF-κB signaling drives part of the toxicity of microglia and is one of the drivers of the disease progression phase.

It would have been interesting to conduct single-cell profiling of all MNs prior to initiating the coculture to investigate if there were particular MN subpopulations present with potential selective resilience to degeneration (S versus FR and FF MNs) as 50% of the cells remained after 72 h of coculture with microglia.

It would also have been enticing to conduct longitudinal analysis of the transcriptional dynamics of individual MNs during their coculture with microglia to reveal if particular MNs can withstand the toxicity from microglia due to modulation of cell intrinsic signaling. TNFα can be both toxic and neuroprotective depending on the status of the neuron and NF-κB transcriptional activation can stimulate expression of prosurvival genes that will block apoptosis such as BCL2 and cellular inhibitors of apoptosis proteins. Perhaps it is a lack of TNF receptors 1 and 2 on particular MNs which underlies resilience, or it is mainly a maturity issue and lack of downstream signaling of NF-κB, but this remains to be investigated.

## 6. Peripheral Macrophages

Immune activation occurs early on in ALS in the peripheral nervous system, with marked infiltration of peripheral macrophages along motor nerves, as shown in mutant SOD1 ALS mice, both fast progressing mice harboring the SOD1^G93A^ mutation and slowly progressing mice harboring the SOD1^G37R^ mutation [170,173]. This finding was also replicated in ALS patients, sporadic and familial with *SOD1* or *C9ORF72* mutations [170]. Macrophages infiltration was followed by immunoglobin deposits and complement activation [173]. The infiltration appears specific to motor nerves as ventral roots showed extensive infiltration while dorsal roots containing the axons of sensory neurons were devoid of macrophages. This specificity in pathology is likely due to motor axons degenerating and releasing factors such as ATP that recruit macrophages along the nerves, while sensory neurons are left intact in ALS.

Chiot and colleagues elegantly demonstrated in the SOD1^G93A^ mouse model that replacing peripheral macrophages with wild-type macrophages had no effect on disease severity. However, replacing them with macrophages in which either wild-type SOD1 was overexpressed or where *Nox2* had been knocked down, which in both cases would result in reduced neurotoxic reactive oxygen species, attenuated both peripheral and central pathology in the mice, without any major recruitment of the peripheral macrophages into the CNS, and improved the survival of the mice [170]. The described activation of central inflammation through the periphery in a pivotal finding which further solidifies the importance of peripheral pathology in driving certain disease stages in ALS. If damage to motor axons is driving the macrophage infiltration, as it seems to, then it is expected that the relative vulnerability of particular motor axons would regulate the level of macrophage infiltration. It has not yet been investigated if there is a temporal difference in recruitment of macrophages to nerves composed mainly of vulnerable FF motor axons versus more resilient S motor axons, and, for that matter, to the highly resilient oculomotor nerve. It would be expected that resilient motor axons would show a low level of infiltration and, thus, no activation of neuroinflammation in the corresponding central nervous system areas.

## 7. Oligodendrocytes

Oligodendrocytes are the myelinating glial cells of the CNS. Oligodendrocytes also provide energy in the form of lactate to axons through monocarboxylate transporters (MCTs). It has been shown that MCT1 (SLC16A1) is highly expressed in oligodendrocytes [174].

MCT1 is also present on MNs and other neuron types in the CNS, as well as on sensory neurons in the dorsal root ganglia, as evidenced by in situ hybridization experiments in the Allen Brain Atlas (mouse.brain-map.org, accessed on June 15 2022). General knock down of MCT1 in vitro and in vivo reduced MN survival. It would have been interesting to perform transcriptional analysis of individual MNs prior to and after MCT1 reduction to see the individual responses as only a portion of MNs were lost. Moreover, it is not clear if these were FF or S or both. Furthermore, MCT1^+/−^ mice developed axonopathy without disturbing the myelination of axons [174]. Thus, functionally, MCT1 would appear more pivotal to MNs than to oligodendrocytes. It has been reported that MCT1 expression is decreased in the spinal cords of SOD1^G93A^ mice as a function of disease and in ALS patients [174,175]. It remains to be further investigated if this decrease is mainly due to loss of MNs expressing MCT1 or modulation of MCT1 expression on glial cells. It also remains to be investigated if there is a differential expression and regulation of MCT1 on particular MN subpopulations and oligodendrocyte populations in health and disease and if lactate dependence could play a role in selective vulnerability. Interestingly, overexpression of MCT1, specifically in oligodendrocytes, and particularly in white matter oligodendrocytes, in SOD1^G93A^ mice using an AAV9 vector did not improve MN loss or the survival of the mice [175]. It remains to be investigated if a broader overexpression targeting multiple cell types could give a beneficial effect.

Ferraiuolo and colleagues showed that mouse MNs cocultured with ALS-oligodendrocytes derived from either mSOD1 mice or ALS patients had lower survival than MNs cocultured onto control oligodendrocytes [176]. *C9ORF72* mutant, sporadic ALS, as well as *SOD1* and *TARDBP* mutant oligodendrocytes all resulted in reduced MN survival and so did conditioned media from all lines, indicating that it was secreted factors that mediated the effect. All oligo lines, except *C9ORF72* mutants, showed reduced lactate levels in the media and conditioned-media-induced MN death was associated with decreased lactate production. Thus, there appears to be mutation-specific modulations in oligodendrocytes. Notably, toxicity in coculture with oligodendrocytes appeared to be lactate-independent and is likely mainly mediated by cell–cell contact [176].

Not only neurons are targets in ALS. In fact, there appears to be an increased degeneration of oligodendrocytes in the spinal cord of SOD1^G93A^ mice in combination with an enhanced generation of NG2^+^ cells and oligodendrocytes. The newly generated oligodendrocytes failed to remyelinate axons, reminiscent of multiple sclerosis [177]. Oligodendrocytes were shown to also be lost in ALS patients combined with an increase in NG2 immunoreactivity in the motor cortex and spinal cord [177]. Excision of mutant SOD1^G37R^ from NG2^+^ cells prolonged the life span of the mice by delaying the onset and early progression of the disease.

Interestingly, MNs are also degenerating in multiple sclerosis [178], linking the two diseases and indicating that MN death can be driven through independent mechanisms. A cross-disease analysis of oligodendrocytes and MNs in ALS and MS may give important clues to MN vulnerability and oligodendrocyte dysfunction.

There is yet no data linking oligodendrocyte heterogeneity and selective vulnerability in ALS, but there is a diversity among mature oligodendrocytes [179]. It was recently shown that there is a spatial preference of particular oligodendrocyte populations and that these show distinct responses to injury [180]. It will be an important task to elucidate if there is a differential response among oligodendrocytes to ALS and if particular subpopulations could be used as therapeutic targets to preserve MNs.

## 8. Schwann Cells

Schwann cells are the primary glial cells of the peripheral nerves and encompass both myelin generating and non-myelinating cells. They share a developmental origin in neural crest and progress from SOX10^+^ GAP43^+^ precursors (at E14 in mouse) into immature Schwann cells marked by expression of SOX10, p75, S100, EGR2 (KROX20), and OCT6 (at E17 in mouse). They then separate into the myelin generating lineage that ensheath somatic motor nerves or the non-myelinating lineage of the somatosensory and autonomic nerves, see review [181]. The myelin-generating Schwann cells (in mouse at birth) are demarked by expression of SOX10, EGR2, S100, and MPZ (myelin protein zero). In contrast to oligodendrocytes, a Schwann cell only myelinates one axon, covering about 100 µm of the length of the axon in mice or 1 mm in humans. Thus, a 50 cm long motor axon in a human would be covered by ca. 500 individual Schwann cells.

The biology of Schwann cells is tied to MNs and vice versa. Not only is their interaction crucial for their mutual development and survival, neuron-derived factors guide the differentiation of Schwann cells along the axon [182] while Schwann cells allow axonal maturation. Schwann cells also greatly facilitate the regeneration of axons and remyelination following injury [183]. Schwann cell-derived neuregulins also enhance local motor axon sprouting [184,185]. Although alterations in myelin sheets in the peripheral nerves are observed in ALS patients, these are generally considered consequences of motor axon degeneration [186] and regeneration [187] and not attributed to Schwann cell dysfunction. Schwann cells appear intrinsically protected from buildup of mutant SOD1 toxicity in mouse models of ALS. Expression of mutant SOD1^G93A^ protein in MPZ^+^ Schwann cells, specifically, yielded mice identical to controls with no changes in locomotion, neuronal loss, or axonal degeneration [188]. This might not be surprising given that afferent sensory fibers are spared in the same peripheral nerve where motor axons are lost.

However, Schwann cells may play a role in the disease that is more subtle. The selective reduction (with 70%) of mutant SOD1^G37R^ protein, that retains its dismutase activity, within Schwann cells (myelinating and non-myelinating) did not alter disease onset, but surprisingly exacerbated disease progression. This would indicate that a high level of dismutase activity in Schwann cells is protective in ALS, as has been shown for wild-type SOD1 in injury [189,190,191], presumably by reducing toxic superoxide anions [192]. If this was the case, then the deletion of an enzymatically dead mutant SOD1 from Schwann cells would be predicted to have no effect on life span. However, depletion of mutant SOD1^G85R^, which is lacking dismutase activity, from Schwann cells delayed disease onset and slowed progression slightly [193]. So, what conclusions can we draw from this? In the study by Lobsiger and colleagues in which SOD1^G37R^ was depleted from Schwann cells, there was a concomitant 50% decrease in the neurotrophic factor IGF1 (insulin-like growth factor I) in sciatic nerve Schwann cells [192]. The authors speculated that this potentially could affect motor neuron regeneration and survival in ALS, which is highly plausible as IGF1/2 promotes nerve regeneration and motor neuron protection in ALS [90,194]. It remains to be investigated if there was any modulation of *Igf* signaling in the SOD1^G85R^-depleted nerve.

Interestingly, sensory neurons signal peripheral nerve injury through induced nitric oxide and upregulate erythropoietin receptors. Schwann cells sense the nitric oxide through the hypoxia inducible factor (HIF1) and secrete neurotrophic erythropoietin (EPO) that activates axon-protective pathways in the neurons [195]. Future studies are needed to clarify the role of different mutant SOD1 forms in motor axons versus sensory axons and further dissection of the role that SOD1 dismutase activity would play in the execution of the oxidative signal in Schwann cells.

Recently, a distinct non-myelinating perisynaptic Schwann cell was described to participate in the NMJ, reviewed in [196]. Perisynaptic Schwann cells most likely share developmental origin with other peripheral Schwann cells and are characterized by specific expression of NG2 and S100β [197]. A typical NMJ in mouse is covered by two to four perisynaptic Schwann cells. The importance of these cells in maintenance and stability of the NMJ is subject to current studies. There is evidence that perisynaptic Schwann cells contribute to selective reinnervation of particular NMJs. Aberrant function or expression of axonal guidance factors have been implicated in this matter, reviewed in [198]. For example, *Sema3a* is an axonal guidance factor which is specifically increased in the perisynaptic Schwann cells of fast-fatigable muscle fibers in SOD^G93A^ mice [199]. This increased expression of *Sema3a* could bias plasticity during regenerative innervation of vacated NMJs [198]. Furthermore, the perisynaptic Schwann cells express higher levels of Sema3a already at presymptomatic stages compared to symptomatic stages in SOD1^G93A^ mice and, thus, might be involved in the initial retraction of the motor axons at these terminals. The implication of such a repellant axon guidance factor in the disease also supports the notion that miscommunication between vulnerable neurons and perisynaptic Schwann cells in muscle might contribute to the denervation of the NMJ and selective vulnerability.

## 9. Skeletal Muscle

Somatic MNs that are vulnerable to ALS control the skeletal muscles in the body. These are striated muscles that connect to bones via tendons and move the skeleton by their contraction. The striated muscles can be voluntarily controlled in contrast to the smooth and cardiac muscles that are innervated by visceral MNs instead of somatic ones. The selective vulnerability of the motor units occurs at different levels in ALS and is reflected on the side of the muscle as well the MN (Figure 1). For example, the motor units that control eye movement (oculomotor, trochlear, and abducens nerve nuclei) and the pelvic floor responsible for urogenital functions are spared in ALS [14,15]. However, even among the somatic motor units exists a gradient of vulnerability that matches fiber subtypes [20]. In humans, the skeletal muscle fibers are divided into three groups based on metabolism, contractile properties, and other features of the motor unit: type I fibers are slow-twitching, fatigue-resistant, and oxidative; and type IIa fibers are fast-twitching, fatigue-resistant, and metabolic intermediate; whereas type IIx fibers are fast-twitching, fatigable, and glycolytic. The fast-twitch glycolytic fibers, innervated by highly vulnerable FF MNs, become denervated and atrophic before slow-twitch oxidative fibers [20,80].

MN death is strongly correlated with the decline in motor function in SOD1 mouse models [80,200]. Neuropathologically, the denervation and degeneration of NMJs is an early event in ALS disease progression that precedes the degeneration of MN somas in SOD1^G93A^ mice. Already by P30, approximately 40% of NMJs in fast-twitch muscles are denervated at a presymptomatic timepoint, whereas MN somas show only a 20% reduction one month later, by P60 [80,201], and some muscles, such as the medial gastrocnemius in mice, are denervated even earlier than P30 [80,202]. The retraction of motor axons from NMJs in SOD1^G93A^ mice occurs even in MNs that offset apoptosis by a deletion of BAX [202]. Furthermore, there are many more indications that destabilization of the NMJ is a critical step in ALS that occurs before and independent of neuroinflammation in the spinal cord [87,203]. We think, therefore, that early local changes at the distal NMJ elicit the denervation observed in ALS. Because dysfunction in non-neuronal cells can exacerbate MN mortality and disease onset [168,204], we asked what the contributions of the skeletal muscles are to the disease.

Both sporadic and familial ALS cases present skeletal muscle pathology that includes signs of mitochondrial dysfunction [205,206,207,208]. Similarly, transgenic mice have metabolic and functional deficits in skeletal muscles [209,210,211]. Mitochondrial dysfunction in muscle in ALS appears detrimental to the stability of NMJs [212,213,214]. The uncoupling of muscle mitochondria by itself results in denervation, MN degeneration, and, finally, exacerbates disease progression [214].

The particular molecular changes leading up to muscular atrophy have been described and include an increase in reactive oxygen species [215], but also compensatory antioxidative activities [211,216]. Atrophy of the gastrocnemius muscle in SOD1^G93A^ mice is preceded by an increase of proinflammatory cytokines and autophagy markers. Autophagy contributes to decreased myofiber size initially, and caspase activity contributes to atrophy only at late stages following denervation [212,217]. Furthermore, a decrease of PGC1α in skeletal muscles has been reported in SOD1^G93A^ mice [218]. PGC1α is a transcriptional coactivator that targets genes involved in the oxidative phosphorylation and that stimulates mitochondrial biogenesis. It has several isoforms that specify skeletal muscle fiber types. Elevated expression of the PGC1α1 isoform switches the skeletal muscle to slow-oxidative fibers associated with endurance phenotypes, while the alternative isoform PGC1α4 is specifically expressed in exercised muscle and induces hypertrophy in skeletal muscle [219,220]. Thus, the early decrease of PGC1α in skeletal muscles in ALS could represent the fiber type switch from the vulnerable fast-twitch fibers to slow-twitch fibers during regenerative innervation instead of simply the selective loss of fast-fiber types. The expression of PGC1α can offset muscular atrophy and, thus, the muscle functionality is retained longer in SOD1^G93A^ mice, but it does not extend life span [221]. There is also further evidence that energy metabolism in ALS muscles is disturbed. An impairment of the cells’ ability to sense the intracellular energy status by AMPK deletion in mice phenocopies motor dysfunction reminiscent of SOD1 mutations [222] and a high-fat energy-rich diet is beneficial in mutant SOD1 mice [210]. A number of studies have modulated SOD1^G93A^ expression in skeletal muscle specifically to understand the role of muscle in ALS. The selective expression of SOD1^G93A^ in skeletal muscle, under the myosin light chain promoter, elicited progressive muscular atrophy, muscular weakness, and mitochondrial dysfunction, but did not trigger degeneration of spinal MNs [212]. Furthermore, the suppression of mutant SOD1 in muscle alone in another study was not sufficient to improve grip strength or prolong the survival of SOD1^G37R^ mice. Here, the suppression of mutant SOD1 was estimated to be about 50% [223]. Such a decrease of mutant SOD1 in microglia is sufficient to substantially delay disease progression [168], and it is, therefore, entirely possible that muscle is not a driver of disease in ALS. However, these experiments do not rule out that mutant SOD1 exerts toxicity onto muscle at very low levels so that it needs to be completely removed for muscle to impact MNs, that themselves harbor high amounts of mutant SOD1, in a beneficial manner. Our analysis of NMJ pathology in SOD1^G93A^ mice clearly shows that pathology starts presynaptically and is subsequently followed by pathology on the postsynaptic muscle side [12]. These results point to MNs as drivers of disease, rather than muscle, but does not rule out that MNs are dependent on myokines that no longer are provided in ALS due to muscle dysfunction and that muscle therefore plays a pivotal role. It seems that muscle alone does not drive disease and neither that rescue of muscle alone can stop the disease, but the final experiments needed to make finite conclusions on this matter remain to be conducted. The importance of muscle-secreted neurotrophic support to NMJ stability and MNs survival is highlighted by studies in which muscle-targeted treatments with neurotrophic factors such as VEGF and BDNF could improve life span and motor function in SOD1^G93A^ mice [224,225,226]. We have previously shown that overexpression of PGC1α1 in primary myotubes increased NMJ formation and size in vitro. PGC1α1 increased the secretion of neurturin from muscle, which was necessary and sufficient for the effects on NMJ formation [227]. This demonstrates that a myokine controls NMJ formation. Muscle-specific neurturin overexpression in mice resulted in a shift of muscle to a slow oxidative type with improved mitochondrial function and also promoted a shift in MN identity towards a slow (S) MN type through retrograde signaling, coupling the characteristics of muscle and MN [228]. This shows for the first time that a muscle-derived neurotrophic factor can reprogram MN somas and that neurturin has therapeutic potential. It has not yet been documented if neurturin is regulated during ALS, but it is known that PGC1α is a disease modifier in men in ALS and that deficiency in full-length PGC1α leads to an earlier age of onset and shortened survival. This was, in part, thought to be mediated by the resulting lower levels of VEGFA [229]. However, as neurturin appears to mediate the majority of the effects of PGC1α1, it may be that it is the subsequent loss of neurturin that renders the loss of PGC1α detrimental in ALS.

Primary myoblasts isolated from ALS patients have a reduced ability to form myofibers and display fusion defects [230]. These pathological changes may be due to an intrinsic deficit rather than an exhaustion of the regenerative pool of satellite cells. Some of the evidence for this comes from careful comparisons of satellite cells in resilient extraocular muscles and vulnerable leg muscles. It has been speculated that satellite cells partake in the extraordinary resilience of extraocular muscles in ALS. However, careful investigation by Lindström and colleagues showed that only a small portion of the extraocular muscles, close to the tendon, maintains a pool of satellite cells, with a majority of the muscles containing low numbers of satellite cells that were in a dormant (rather than activated) state, at similar numbers to adult limb muscles [231]. Analysis of extraocular muscles from ALS patients revealed similar results and showed that while there were varying numbers of satellite cells in limb muscles of ALS patients, they did not seem to wear out or decrease in numbers but were within the normal variation of elderly individuals [232,233]. Thus, it may be that intrinsic properties of satellite cells and their ability to form fibers affect their vulnerability in ALS rather than the numbers of cells that are there.

## 10. Perivascular Fibroblasts in the Blood–Brain Barrier

Blood–brain barrier disruption has been associated with multiple neurodegenerative diseases including ALS, reviewed in [234]. Several mutant SOD1 mouse models display early vascular changes such that the blood–brain or blood–spinal cord barrier break before onset of motor phenotypes [235,236]. A recent study showed that genes with known preferential expression in perivascular fibroblasts (as inferred from single-cell RNA sequencing data) were upregulated in bulk RNA sequencing data from postmortem spinal cord tissue from sporadic ALS patients. Analysis of early symptomatic stages in SOD1^G93A^ mice showed that such markers were upregulated already at 4 weeks of age and even stronger at 8 weeks, a time point when NMJ disruption has started to initiate in these mice and prior to when astrocyte and microglia activation occur. Perivascular markers were also increased in TARDBP^Q331K^ mice at relatively early disease stages. This would indicate that the vasculature is an early pathological target in ALS, preceding inflammation. Analysis of a large number of plasma samples from sporadic ALS patients with recent ALS diagnosis showed a correlation between elevated levels of the protein marker SPP1 for perivascular fibroblasts and a more aggressive disease progression and thus shorter survival from onset of disease, which indicates a connection between the vasculature and sporadic ALS [237].

However, a recent study that tried to connect blood–brain barrier disruption and disease severity using the ratio of CSF to serum albumin levels as a marker failed to associate disease aggressiveness with disruption of the blood–brain barrier except for in patients with limb-onset [238]. Further, the patterns of TDP43 accumulation and blood–spinal cord barrier leakage do not match up in the spinal cord of the end-stage ALS patients [239]. This suggests that the TDP43 pathology and blood–spinal cord barrier breakdown may be independent events in ALS. It remains to be investigated if there is a heterogeneity among perivascular fibroblasts and/or diversity in their response to ALS that could contribute to selective vulnerability. 

## 11. Conclusions

The progressive dysfunction and loss of motor neurons in ALS results in the decay of motor function. This degeneration involves both motor neuron intrinsic changes and several other cell types across the motor system, from the motor cortex to the neuromuscular junctions (Figure 1). The different degrees of vulnerability across somatic motor neurons, although not fully explained yet, offers the opportunity to uncover protective mechanisms as well as determinants of the degenerative processes. Recent studies have greatly advanced our ability to distinguish the different populations of somatic neurons as well as the diversity among other neuron groups and glial cells, and we expect that these will accelerate the study of the disease in the near future.

ALS, like many human neurodegenerative diseases, is difficult to interrogate mechanistically and we are limited by insight into the human nervous system. Our conclusions rest, therefore, on studies in suitable model systems that are validated or corroborated in humans wherever possible. Several mouse models based on other ALS-causative genes have been created in the last decades reviewed in [240,241,242] along with models in other organisms reviewed in [243]. Still, our current understanding of the cellular and molecular mechanisms that define and shape ALS is based, at large, on the careful characterization of mutant SOD1 rodent models, especially at early stages of the disease. The detailed cross-comparison of several diverse models will aid the identification of shared pathological mechanisms as well as unique features of individual forms of ALS. Another caveat is the current lack of complex models of sporadic ALS that constitutes the majority of cases due to our missing insight into its causation. Animal models that recapitulate TARDBP pathology are of particular interest because it represents a hallmark in the majority of either familial or sporadic ALS.

Given the multiple sites of pathology in the body during the course of ALS and the many cell types involved, it is an important challenge to establish timelines of the cellular and molecular changes and causal links between alterations where possible. 

## Figures and Tables

**Figure 1 biology-11-01191-f001:**
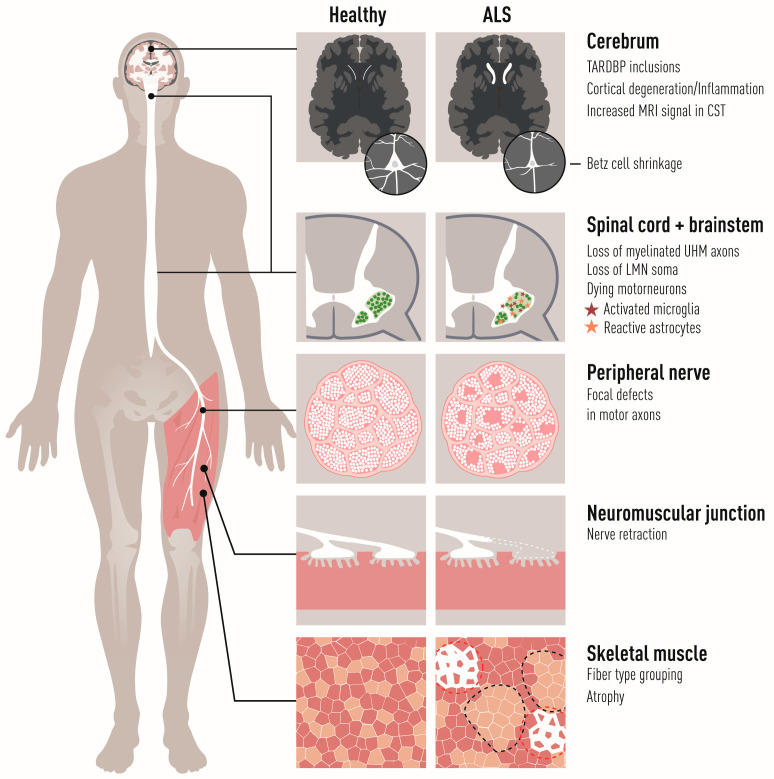
**The pathology of ALS encompasses all levels of the voluntary neuromuscular system and involves multiple cell types.** The primary targets of ALS are motor neurons in the cortex, brainstem, and spinal cord. Neuropathological characteristics of most ALS cases are TDP43 inclusions in both neurons and glial cells. Neuroinflammation is raging throughout the nervous system with activated microglia and astrocytes seen wherever degeneration is occurring. The loss of upper and lower motor neurons is further evident from axonal degeneration and demyelination of the corticospinal tract as well as of the peripheral spinal nerves innervating skeletal muscles [2,3]. Denervation of neuromuscular junctions on affected skeletal muscle fibers have begun prior to clinical onset of any apparent motor deficit. While FF motor axons retract and degenerate, S motor neurons sprout and innervate endplates vacated by FF motor neurons. This new innervation results in fiber type switching and apparent grouping of muscle fiber types followed by atrophy of permanently denervated muscle fibers [4,5,6].

**Figure 2 biology-11-01191-f002:**
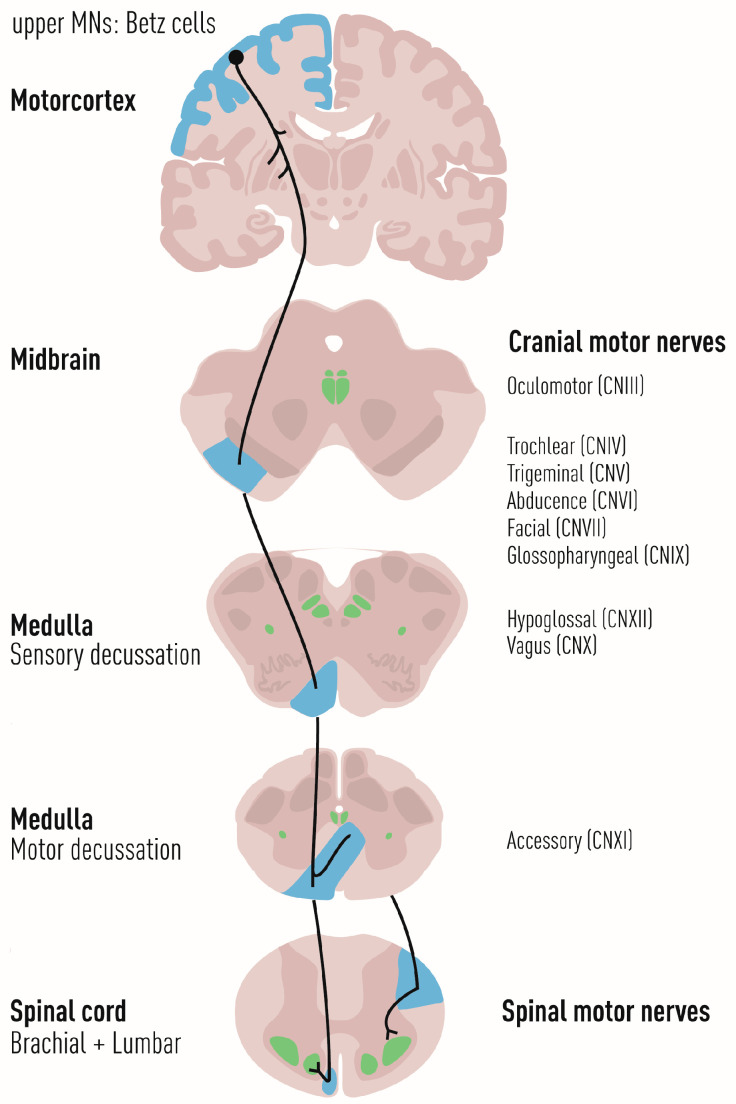
**The upper motor system in humans and its connectivity with the lower motor system.** Upper motor neurons are large pyramidal cells in the motor cortex, that are called Betz cells (blue). Their axons project through the corticospinal tract and innervate the nuclei of different lower motor neurons in the brainstem and spinal cord (green). Upper motor neurons are vulnerable to degeneration in ALS. Lower motor neurons are responsible for voluntary control of skeletal muscles and are vulnerable in ALS to different degrees.

**Figure 3 biology-11-01191-f003:**
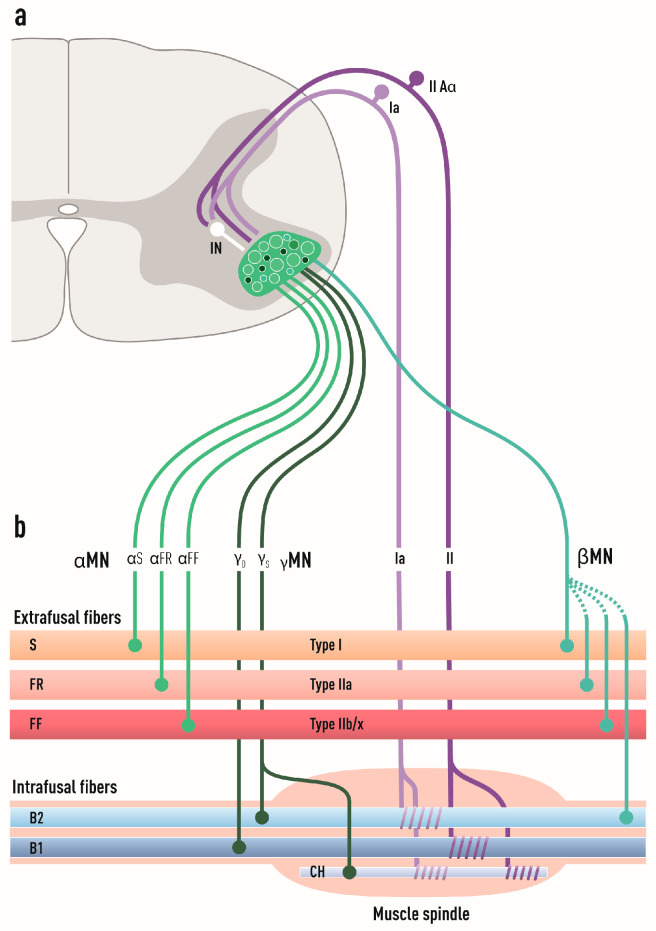
**The subdivision of lower somatic motor neurons and their connectivity with different muscle fiber types.** (**a**) Lower motor neurons located in the ventral horn of the spinal cord innervate skeletal muscle fibers. They receive modulating signals from interneurons and also integrate inhibitory signals from afferent sensory neurons (Ia and II group IIAα) through the skeletal muscle that they both innervate. Although motor neuron somas are segregated according to muscle group in the motor column, the different motor neuron subtypes (alpha, beta, gamma) are intermingled. (**b**) Alpha motor neurons innervate extrafusal skeletal muscle fibers, which generate contractile forces. Alpha motor neurons are subdivided into fast-twitch fast fatigable (FF), fast-twitch fatigue-resistant (FR), and slow-twitch (S), which innervate type IIb/x, type IIa, and type I muscle fibers, respectively. Gamma motor neurons that are coactivated together with alpha motor neurons innervate intrafusal fibers that form stretch receptors in the form of muscle spindles and create a parallel pull of extra- and intrafusal fibers. The stretching of the spindle is detected by sensory neurons wrapping around the intrafusal fibers. Gamma motor neurons form two subtypes (dynamic and static) innervating intrafusal nuclear bag fibers (B1, B2) and nuclear chain fibers (CH). The dynamic units are required for quick adjustments to muscle tonus, whereas the static units are mainly involved in maintaining muscle tonus while posturing. Beta motor neurons innervate the muscle spindles and also have collaterals to the extrafusal fibers.

**Figure 4 biology-11-01191-f004:**
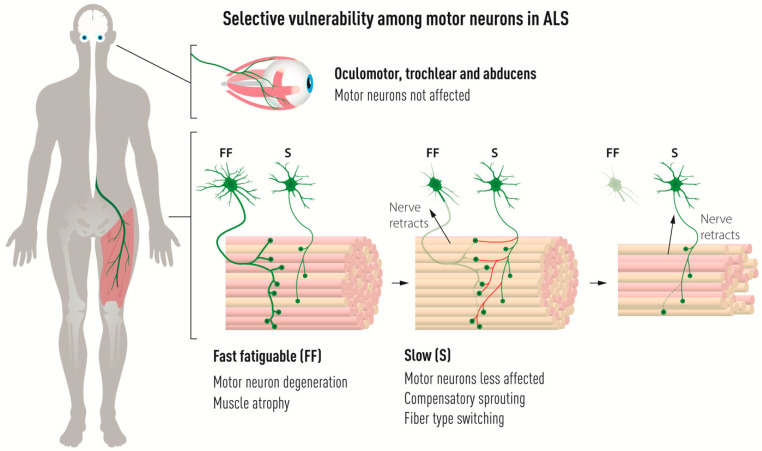
**Selective vulnerability among somatic motor neurons in ALS.** In ALS, somatic motor neurons of cranial motor neurons of the oculomotor, trochlear, and abducens nuclei, which regulate eye movement, remain unaffected until the end-stage of the disease. Among spinal motor neurons, there is a differential vulnerability with fast fatigable (FF) motor neurons being highly vulnerable and degenerating before symptoms onset, with resulting muscle atrophy. Slow (S) motor neurons are affected later in the disease and compensate for the loss of FF motor neurons by sprouting and innervating vacated endplates that were previously occupied by FF motor neurons. This results in fiber type switching of muscle from fast to slow. Eventually, S motor neurons are also affected and their NMJs with muscle are disrupted and their axons retract from muscle with resulting muscle weakness and atrophy.

**Figure 5 biology-11-01191-f005:**
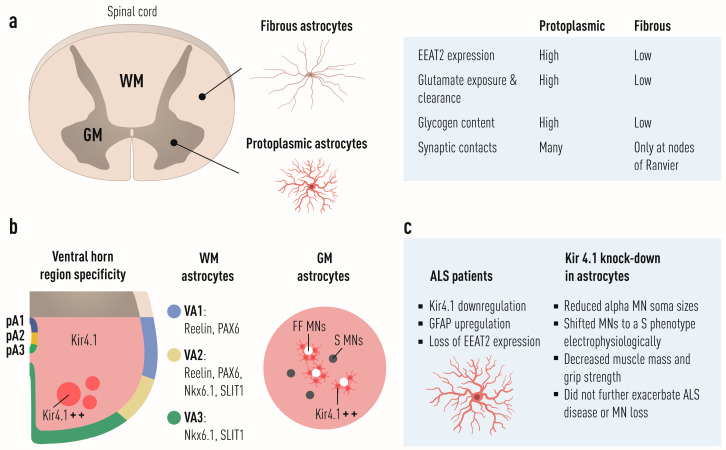
**Dorsoventral astrocyte diversity in the spinal cord and implications for differential spinal motor neuron vulnerability in ALS.** (**a**) Astrocytes are subdivided into fibrous, in the white matter (WM), and protoplasmic, in the gray matter (GM). These differ in their EEAT2 expression, glutamate exposure and clearance, glycogen content, and synaptic contacts. (**b**) Astrocytes within the WM show regional specificity and arise from three progenitor domains, p1, p2, and p3. These subtypes can be identified on the basis of their expression of PAX6, NKX6.1, and the cell surface markers, reelin and SLIT1. Specifically, dorsally located VA1 astrocytes, derived from the p1 domain, express PAX6 and reelin; ventrally located VA3 astrocytes, derived from the p3 domain, express NKX6.1 and SLIT1,]; and intermediate WM-located VA2 astrocytes, derived from the p2 domain, express PAX6, NKX6.1, reelin, and SLIT1 [136]. GM astrocytes also show regional specificity with clear differences in KIR4.1 (KCNJ10) expression, with ventral horn astrocytes exhibiting higher expression of KIR4.1 than dorsal horn astrocytes. Within the ventral horn, WM astrocytes surrounding FF MNs show the highest level of KIR4.1 (KIR4.1^++^) and those surrounding S MNs show a lower level of KIR4.1. (**c**) ALS patient astrocytes show decreased expression of KIR4.1, GFAP upregulation and loss of EEAT2 expression. Removal of KIR4.1 from astrocytes in mice affected αMN soma sizes and shifted electrophysiological properties to a more slow-like MN phenotype with an accompanying decrease in muscle mass and grip strength. Overexpression of KIR4.1 specifically in astrocytes was sufficient to increase the size of both FF and S MNs. However, loss of KIR4.1 in astrocytes in the SOD1^G93A^ mouse did not further exacerbate MN death.

## Data Availability

Not applicable.

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
