# Peer review of "The Cell Autonomous and Non-Cell Autonomous Aspects of Neuronal Vulnerability and Resilience in Amyotrophic Lateral Sclerosis"

_biology, 2022, doi:10.3390/biology11081191_

Round 1

Reviewer 1 Report

In this review, Schweingruber and Hedlund presented a comprehensive review on the cell autonomous and non-cell autonomous aspects of ALS and the selective vulnerability and resilience of neuronal and non-neuronal cell types. The authors sought to understand why some neuronal populations are particularly prone to degeneration and some neurons are resilient (ocular motor neurons). The authors also surveyed a multitude of non-neuronal cells and how they contribute to ALS progression. The review is very comprehensive and high quality and I think it would benefit the ALS researchers. I just have several minor clarifications. 

Major criticisms

1.    Line 91: NMJ has been defined in line 39. Some acronyms have been defined but were refined subsequently in the paper. 

2.    Line 163: “LCM identity as removal of this…” I assume it is LMC

3.    Line 415: NEFH used only once in the paper. Probably don’t need to define the acronym.

4.    The gene nomenclature and style are at times not consistent (i.e.: Italics, full capitalization). e.g.: line 441-444. Gdf15, GDF15 would mean Gdf5 mRNA and protein respectively.

5.    The interneurons section (line 456-522). Are the authors combining two different types of the interneurons (cortical inhibitory neurons and spinal interneurons) into one section? Would it be beneficial to separate them into two sections given that they are quite distinct (both anatomical location and function wise)?

6.    For skeletal muscle section (line 997-1028), the first two paragraphs are related to the motor neurons and not the skeletal muscle itself. Can the title be renamed to something more inclusive?

Minor criticisms

1.    There are multiple formatting errors which I assume is computer generate due to pdf conversion (line 287-301, 636-675, etc.).

Author Response

In this review, Schweingruber and Hedlund presented a comprehensive review on the cell autonomous and non-cell autonomous aspects of ALS and the selective vulnerability and resilience of neuronal and non-neuronal cell types. The authors sought to understand why some neuronal populations are particularly prone to degeneration and some neurons are resilient (ocular motor neurons). The authors also surveyed a multitude of non-neuronal cells and how they contribute to ALS progression. The review is very comprehensive and high quality and I think it would benefit the ALS researchers. I just have several minor clarifications.

Response: We thank the reviewer for the overall positive assessment and the constructive points raised below. They helped us improving the manuscript.

Major criticisms

  1. Line 91: NMJ has been defined in line 39. Some acronyms have been defined but were refined subsequently in the paper.

Response: The definition of the acronym is now restricted to the first instance.

  1. Line 163: “LCM identity as removal of this…” I assume it is LMC

Response: Thank you for finding this misleading error. It has been correct to the lateral motor column (LMC) in the revised manuscirpt.

  1. Line 415: NEFH used only once in the paper. Probably don’t need to define the acronym.

Response: We have removed the acronym in the revised manuscript.

  1. The gene nomenclature and style are at times not consistent (i.e.: Italics, full capitalization). e.g.: line 441-444. Gdf15, GDF15 would mean Gdf5 mRNA and protein respectively.

Response: We have updated the gene nomenclature throughout the revised manuscript. In short for mouse: Isl1(gene and transcript), ISL1 (protein); for human: ISL1 (gene and transcript) and ISL1 (protein) and removed hyphenation. In few instances we have chosen to provide alternative names, especially if they are in common use.

Consistently, we have chosen to call the TARDBP protein “TDP43” in the manuscript because it more common in the literature, especially in phrases such as “TDP43 pathology”. However, when referring to the gene directly, we also use its proper gene name TARDBP.

  1. The interneurons section (line 456-522). Are the authors combining two different types of the interneurons (cortical inhibitory neurons and spinal interneurons) into one section? Would it be beneficial to separate them into two sections given that they are quite distinct (both anatomical location and function wise)?

Response: Indeed, we discuss interneurons in the cortex and spinal cord in this section of the manuscript. We agree with the reviewer that it is important to keep them apart in regards to anatomy and function. In the original manuscript we have identified these groups as cortical and spinal interneurons, and have discussed them in separate paragraphs. In order to more clearly separate them, we have now included appropriate section headings (line 470 and 485).

  1. For skeletal muscle section (line 997-1028), the first two paragraphs are related to the motor neurons and not the skeletal muscle itself. Can the title be renamed to something more inclusive?

Response: We understand that the two paragraphs are a wordy segue to the actual changes observed in skeletal muscle itself. However, these paragraphs give essential context to the skeletal muscle pathology observed in ALS at large. We disagree with them not being related to the skeletal muscle. In our opinion, these two paragraphs rather reflect on the intimate relation of skeletal muscle to the motor neurons.

In short, the first paragraph is meant to establish vulnerable muscle groups and fiber types, which is a function of the decline of the motor unit (muscle and neuron). Naturally there is some redundancy to earlier paragraphs introducing motor neuron subtypes. The second paragraph highlights that the disconnection of motor neuron and skeletal muscle preceeds the overt degeneration of either. In our stated opinion, it is local changes at the NMJ that elicit the denervation and thus we ask about the contributions of the skeletal muscle to it.

Minor criticisms

  1. There are multiple formatting errors which I assume is computer generate due to pdf conversion (line 287-301, 636-675, etc.).

Response: Thank you pointing out several formatting errors in the manuscript. We have corrected them and several more that we have found while preparing the revised manuscript. Given their number and minor nature, we would like to refer you to the revised manuscript in track change mode instead of enumerating them here.

Reviewer 2 Report

The review article titled “The cell autonomous and non-cell autonomous aspects of neuronal vulnerability and resilience in amyotrophic lateral sclerosis” by Hedlund and Schweingruber describes the role of different types of neuronal cells and non-neuronal cells to ALS. Intrinsic and extrinsic mechanisms are discussed where appropriate.

This is a highly comprehensive review and gives a through overview of the contributions of different neuronal cell types in much detail and non-neuronal cell types to ALS pathogenesis.

The figures are helpful and effectively illustrate major points outlined in the text.

The text is mostly clear and easy to follow.

Only, extensive language editing (punctuation, incomplete sentences) is required.

Also, certain statements require the addition of literature references.

The authors mention gaps of knowledge and often suggest futureexperiments to close these gaps.

Some limitations of the published research reviewed here should be emphasized, e.g., most experimental data originates solely from SOD1 mutant mouse models and not many other ALS mouse models.

I think the Conclusion should be expanded. The authors could comment on a hierachy of contributions of different cell types and processes (central, highly important vs. less important; early in ALS pathogenesis vs. late; primary ALS effects vs. secondary effects).

Author Response

Comments and Suggestions for Authors

The review article titled “The cell autonomous and non-cell autonomous aspects of neuronal vulnerability and resilience in amyotrophic lateral sclerosis” by Hedlund and Schweingruber describes the role of different types of neuronal cells and non-neuronal cells to ALS. Intrinsic and extrinsic mechanisms are discussed where appropriate.

This is a highly comprehensive review and gives a through overview of the contributions of different neuronal cell types in much detail and non-neuronal cell types to ALS pathogenesis.

The figures are helpful and effectively illustrate major points outlined in the text.

The text is mostly clear and easy to follow.

Only, extensive language editing (punctuation, incomplete sentences) is required.

Also, certain statements require the addition of literature references.

The authors mention gaps of knowledge and often suggest futureexperiments to close these gaps.

Response: We thank the reviewer for the detailed and positive assessement.

Some limitations of the published research reviewed here should be emphasized, e.g., most experimental data originates solely from SOD1 mutant mouse models and not many other ALS mouse models.

I think the Conclusion should be expanded. The authors could comment on a hierachy of contributions of different cell types and processes (central, highly important vs. less important; early in ALS pathogenesis vs. late; primary ALS effects vs. secondary effects).

Response: We have expanded the conclusion of the manuscript to reflect on the last two points raised by the reviewer, namely the predominance of the SOD1 mutant mouse models and the potential hierarchy in the contributions of the different cell types. The relevant passages read now:

“ALS like many human neurodegenerative diseases is difficult to interrogate mechanistically and we are limited by insight into the human nervous system. Our conclusions rest therefore on studies in suitable model systems that are validated or corroborated in humans wherever possible. Several mouse models based on other ALS causative genes have been created in the last decades [reviewed in 240–242] along models in other organism [reviewed in 243]. Still, our current understanding of the cellular and molecular mechanism that define and shape ALS is based at large on the careful characterization of mutant SOD1 rodent models, especially at early stages of the disease. The detailed cross-comparison of several diverse models will aid the identification of shared pathological mechanisms as well as unique features of individual forms of ALS. Another caveat is the current lack of complex models of sporadic ALS that constitutes the majority of cases due to our missing insight into its causation. Animal models that recapitulate TARDBP pathology are of particular interest because it represents a hallmark in the majority of either familial or sporadic ALS.

Given the multiple sites of pathology in the body during the course of ALS and the many cell types involved, it is an important challenge to establish timelines of the cellular and molecular changes and causal links between alterations where possible. “

Reviewer 3 Report

In this review article entitled "The cell autonomous and non-cell autonomous aspects of neuronal vulnerability and resilience in amyotrophic lateral sclerosis", Schweingruber and Hedlund reviewed current literature on cell autonomous and non-cell autonomous initiation and progression of motor neurons (MNs) degeneration in amyotrophic lateral sclerosis (ALS).

I found the article well written and organized. It is also easy to follow and previous published evidences are logically presented. I do not have major concerns on the manuscript.

I just would ask to the authors to provide a revised version of the manuscript with new figure 1, figure 4 and figure 5. In the current version is barely readable and it is not possible to assess these figures.

Author Response

In this review article entitled "The cell autonomous and non-cell autonomous aspects of neuronal vulnerability and resilience in amyotrophic lateral sclerosis", Schweingruber and Hedlund reviewed current literature on cell autonomous and non-cell autonomous initiation and progression of motor neurons (MNs) degeneration in amyotrophic lateral sclerosis (ALS).

I found the article well written and organized. It is also easy to follow and previous published evidences are logically presented. I do not have major concerns on the manuscript.

Response: We thank the reviewer for the overall positive assessement.

I just would ask to the authors to provide a revised version of the manuscript with new figure 1, figure 4 and figure 5. In the current version is barely readable and it is not possible to assess these figures.

Response: We apologize for the poor image quality, which is due to a conversion error. We have exchanged the figures at high resolution in the revised manuscript and we look forward to your assessment.

Round 2

Reviewer 2 Report

The revised version of the manuscript is,in my opinion, ready for publication.

I am certain that this review article will be a useful resource for ALS researchers and beyond.